# How does digital transformation affect the ESG performance of Chinese manufacturing state-owned enterprises?—Based on the mediating mechanism of dynamic capabilities and the moderating mechanism of the institutional environment

**Xin Jin, Yizhu Wu** *

Shanghai Maritime University, Shanghai, China

* wyz344910551@163.com

## Abstract

Against the background of sustainable development policies, the ESG performance of Chinese manufacturing enterprises is still generally poor. As the leading enterprises in the manufacturing industry, state-owned enterprises should take the lead in responding to the national call for sustainable development and actively explore the path to improve their ESG performance. This study aims to explore whether and how state-owned manufacturing enterprises can improve their poor ESG performance through digital transformation in the digital economy. This study takes Shanghai and Shenzhen A-share state-owned listed manufacturing enterprises as the research sample and constructs an unbalanced panel. OLS regression analysis is used to empirically test the impact of digital transformation on the ESG performance of the sample firms. Further attempts are made to discuss the influence mechanism of digital transformation from the perspectives of dynamic capabilities and the institutional environment through stepwise and hierarchical regression methods, respectively. The study shows that, firstly, digital transformation is an important influencing factor in promoting the improvement of enterprises' ESG performance, and at the same time, there are significant structural differences in this influence. Second, under the dynamic capability perspective, digital transformation can improve corporate ESG performance through an absorptive feedback mechanism, matching response mechanism, and innovation efficiency enhancement mechanism. Third, from the perspective of the institutional environment, the informal system has a significant positive moderating effect on the relationship between digital transformation and ESG performance, i.e., the informal system and digital transformation have a synergistic governance effect on corporate ESG performance. The moderating effect of the formal institutional environment on digital transformation and ESG performance is not significant. The findings of the study clarify the controversy over the relationship between digital transformation and ESG performance of manufacturing state-owned enterprises and enrich the research on the influencing factors of corporate ESG performance. It also

**Data Availability Statement:** All relevant data are within the paper and its Supporting information files.

**Funding:** The author(s) received no specific funding for this work.

**Competing interests:** The authors have declared that no competing interests exist.

provides a theoretical foundation and empirical evidence for manufacturing SOEs to improve ESG performance and lead to sustainable development.

## 1 Introduction and literature review

The report of the 20th National Congress of the Communist Party of China (CPC) pointed out that the implementation of the new development concept is the road to China's development and growth in the new era. ESG (Environmental, Social, and Governance) advocates the protection of the ecological environment, the fulfillment of social responsibility, and the improvement of governance, emphasizing the focus on the long-term value of all relevant stakeholders. It is consistent with China's new development concept of innovation, coordination, greenness, openness, and sharing, and is in line with the top-level strategic plan to achieve the "dual-carbon" goal and improve modernized social governance. ESG, as a value of sustainable and coordinated development that considers economic, environmental, social, and governance benefits, is an investment philosophy that pursues long-term value growth and a comprehensive, concrete, and pragmatic approach to governance [1]. For governments, corporate managers, and investors, ESG has been recognized as an important evaluation criterion for corporate fulfillment of sustainability commitments [2], and the better a company's ESG performance indicates a higher level of balancing environmental protection, responsibility fulfillment, and corporate governance, and a stronger capacity for sustainable corporate development. However, due to China's late start in ESG system construction, the overall performance of Chinese companies in ESG ratings is on the low side. Data from the ESG Research Report on Chinese Listed Companies (2023) shows that only 11.86% of Chinese listed companies are rated A, with the manufacturing industry being one of the industries with high ESG risks. From the perspective of China's manufacturing industry, state-owned enterprises have always accounted for a relatively large proportion of the manufacturing industry, and most of them are dominant enterprises in key areas such as petroleum, chemical industry, energy, and other key areas of high energy consumption and high pollution [3], so there is an urgent need to clarify the factors affecting the ESG performance of SOEs in the manufacturing industry.

From an internal perspective, good management decisions are an important factor influencing firms' ESG performance [4], so most of the existing studies have focused on the role of factors such as board characteristics and executive quality on ESG. For example, Husted and De Sousa (2019) found that board size and independent directors have a positive effect on ESG disclosure, while female board members and CEO duality have a negative effect on ESG disclosure [5]. Borghsi et al. (2014) found that CEO gender and age affect corporate socially responsible investment and that firms with female CEOs and younger CEOs have better ESG performance [6]. In addition, both short-sighted managers and overconfident managers are detrimental to firms' ESG performance [7,8]. From outside the firm, regulatory policies tend to shape corporate ESG behavior. For example, Huang et al. (2023) find that firms with important government customers exhibit higher ESG performance than firms without government customers, pointing out that government procurement as an important public policy can significantly affect the ESG performance of supplier firms [9]. Xue et al. (2023), based on the theory of Chinese politics and business, find that local green finance policies can alleviate financing constraints and significantly improve corporate ESG performance [10].

In recent years, the application of digital technology has become more and more extensive. Reprogrammable functionality and data homogenization of digital technology have changed all aspects of product innovation, process innovation, organizational innovation, and business

model innovation. In the era of the digital economy, the data processing capacity crosses from KB level to PB level, and "ABCD" technologies such as Artificial Intelligence, Blockchain, Cloud Computing, Big Data, etc. are widely used. The extensive use of "ABCD" technologies has gradually changed the fundamentals of the industrial economy [11], resulting in great changes in the environment for enterprise development. Digital transformation has become an important strategic choice for micro-enterprise subjects under the high-quality development of the digital economy in the new era.

Many studies have shown that digital transformation helps enterprises reduce costs increase efficiency, and improve financial performance, capital market share, and other economic performance [12,13]. At the same time, it also helps to improve non-economic performance such as corporate social responsibility, human resource management efficiency, and green development capability [14–16]. The UN 2030 Agenda highlights the potential of digital technologies to improve sustainable development performance [17], and there are also studies suggesting that digitalization may have a positive impact on ESG. However, most of the current research on the mechanisms by which digital transformation affects ESG focuses on internal control [18], green innovation [19,20], and financing constraints [2], which is not yet sufficiently comprehensive and in-depth [21].

The resource-based view emphasizes the important values of Valuable, Rare, Imperfect Imitability, and Non-Substitutability of digital resources, but fails to answer the question of where the heterogeneous resources of the enterprise come from and how to use them, and it is also difficult to explain how enterprises can maintain a sustainable competitive advantage in a dynamic environment. Dynamic capability theory, as a development of the resource-based view, can answer the above questions. Dynamic capability is a capability that can change capabilities, empowering digital enterprises to integrate, build, and reconfigure digital resources to cope with rapid changes in the environment, thus enhancing the sustainable competitive advantage of the company [22]. Unfortunately, however, no studies have explored the mechanisms of influence that examine the relationship between digital transformation on firms' ESG performance under the dynamic capability perspective. In addition, institutional theory suggests that in addition to the special resources possessed by firms, the institutional environment based on formal and informal institutions is also an important factor in determining the behavior of micro firms, which is closely related to the ability of firms to achieve competitive advantage [23]. Therefore, in recent years, institutional theory has been widely used in research related to sustainable development, but no study has examined the institutional environment as a weighting factor in the relationship between digital transformation and corporate ESG performance.

In addition, the existing results of the research specifically for state-owned enterprises are relatively limited, and the relationship between digital transformation and the ESG performance of state-owned enterprises has not yet produced a conclusive study. On the one hand, it is believed that enterprise digital transformation is a long-lasting process that requires a lot of resources [7]. Compared with non-state-owned enterprises, state-owned enterprises have greater advantages in resources, talent, policy support, etc., so once state-owned enterprises implement the transformation, their practical results tend to be more excellent. Moreover, due to their unique ownership nature, manufacturing SOEs tend to better implement and enforce national policies [11], so they will more actively respond to the "dual-carbon" policy, take the initiative to assume corporate social responsibility and take the initiative to use digital transformation to improve ESG responsibility performance is relatively stronger. On the other hand, Hajli et al. (2015) found that the effect of enterprise digital transformation on the enhancement of high-quality development of firms is only valid for some of the firms, and a significant number of firms do not benefit from enterprise digital transformation [24]. The market

environment is changing rapidly in the digital economy while manufacturing SOEs are often large in scale and face high sunk costs and structural rigidity in their transformation and upgrading [25]. Moreover, SOEs are relatively backward in terms of innovation ability and organizational management mode, and digital technology is difficult to integrate with their original resources, and their transformation and upgrading face technical barriers and talent gaps [3], resulting in the inability to fully release the digital dividend, and may also have a crowding-out effect on the fulfillment of corporate responsibility.

In general, China's ESG system construction is relatively late, and the ESG performance of the manufacturing industry is generally poor. As leading enterprises in the manufacturing industry, state-owned enterprises (SOEs) should take the lead in responding to the country's call for sustainable development and actively explore paths to improve their ESG performance. In the era of the digital economy, it is worth exploring whether and how state-owned manufacturing enterprises can improve their poor ESG performance through digital transformation. Based on this, this paper constructs an unbalanced panel with state-owned listed enterprises in the manufacturing industry in Shanghai and Shenzhen A-shares as the research sample. OLS regression analysis is used to empirically test the impact of digital transformation on the ESG performance of the sample companies. Further attempts are made to discuss the influence mechanism of digital transformation from the perspectives of dynamic capabilities and the institutional environment through the stepwise method and hierarchical regression method respectively.

The rest of the paper is organized as follows. The Theoretical analysis and research hypothesis section articulates the research hypotheses and discusses the mechanisms by which digital transformation affects ESG performance. The Research design section describes the selected samples, the measurement method of variables, and the corresponding research model. The Hypothesis testing and discussion of results section describes the process and results of the empirical tests. The Discussion section discusses the test results and research contributions. Conclusions and prospects are presented in the Conclusions and enlightenments section.

## 2 Theoretical analysis and research hypothesis

### 2.1 Digital transformation and corporate ESG performance

Under the background of "dual-carbon", the state has gradually introduced a series of supporting policies, and the ESG-related strategies and performance of enterprises have been increasingly emphasized by all walks of life. The political view is that the policy tasks are more often undertaken by state-owned enterprises rather than private enterprises and that SOEs, as leading enterprises in key manufacturing industries, should take the lead in responding to stakeholders' demands, proactively disclosing their ESG status, and improving their own ESG performance. However, traditional state-owned enterprises in the manufacturing industry often face problems such as poor internal and external information communication and low efficiency of technological innovation, which leads to difficulties in collecting and accounting for ESG information, low quality of ESG information disclosure, and unsatisfactory ESG governance effects. In this context, the integration and application of big data, cloud computing, 5G, and other digital technologies to assist enterprises in ESG governance and promote the improvement of corporate ESG performance has become a major trend.

The ESG performance of enterprises is determined by a combination of three factors, including environmental responsibility performance (E), social responsibility performance (S), and corporate governance performance (G). Among them, the environmental responsibility performance (E) of an enterprise reflects the efficiency and effectiveness of an enterprise in continuously improving its pollution prevention and resource utilization through the effective

development of environmental management strategies, and digital transformation can improve the environmental responsibility performance of manufacturing state-owned enterprises from two aspects. On the one hand, digital transformation enhances the sensitivity of state-owned enterprises to environmental issues: with the help of intelligent algorithms, big data, and other technological means, state-owned enterprises can quickly capture social pain points and public environmental problems [26], to identify their deficiencies in environmental responsibility, and adjust their development strategies, such as responding to the "dual-carbon" policy promptly. On the other hand, digital transformation enhances the environmental governance capacity of state-owned enterprises: digital transformation can improve the high input, high consumption, and high pollution status of manufacturing enterprises by reducing the cost of green innovation, improving the green innovation performance of enterprises, and then empowering the green ecological and renewable cycle of manufacturing processes [27], and lowering carbon emissions in the production process. For example, the application of the Internet of Things (IoT) can realize real-time monitoring of ecological changes in the production process and help enterprises to realize carbon footprint measurement, environmental impact assessment, sustainability report, etc., to effectively control the sources of pollution and reduce the waste of resources and pollutant emissions.

The social responsibility performance (S) of enterprises in ESG mainly emphasizes the need for enterprises to strengthen the governance of the relationship between multiple stakeholders, such as shareholders, creditors, employees, customers, and competitors, and to provide them with diversified and comprehensive value in the production and operation process. Based on the stakeholder perspective, CSR can be divided into internal social responsibility related to internal stakeholders and external social responsibility related to external stakeholders [28,29]. Due to the difference in equity, compared with private enterprises, SOEs prefer to take on social responsibility related to internal stakeholders (e.g., shareholders, employees), while paying less attention to the responsibility of external stakeholders (e.g., consumers, suppliers) and so on. The inclusive features of digital technology, such as "accessibility", "openness" and "inclusiveness", can make the contact and communication between SOEs and other external economic entities more extensive and faster, which is beneficial to external stakeholders. and fast, which is conducive to the multiple demands of external stakeholders [26]. Secondly, based on digital technology, SOEs can capture and analyze the value demands of various stakeholders more quickly. The strong innovation orientation of digitalization will in turn empower SOEs to generate new processes and techniques in response to the multiple value propositions of various stakeholders [30]. In addition, research has shown that digital transformation can break down physical resource barriers, improve resource allocation efficiency, and thus enhance corporate economic performance [31], which can also create greater benefits for shareholders and further enhance the internal social responsibility performance of SOEs.

For corporate governance performance (G). Managers of traditional state-owned enterprises tend to be "empirical". Based on the massive data of production and operation, data mining systems and decision support systems can help managers clarify the correlation between things and the probability of managers making irrational decisions based on experience and intuition will be greatly reduced. For middle and low-level leaders or ordinary employees, due to the flattening of the structure of digital enterprises, the speed of internal information transmission and the degree of sharing has increased significantly, and they will have more right to speak and can improve the internal supervision and feedback system of traditional state-owned enterprises [32]. In addition, the application of big data promotes the interconnection of data, greatly improves the information disclosure environment of the capital market, and significantly reduces the cost of information acquisition for external participants such as investment institutions, market service institutions, and news media [33]. State-

owned enterprises will face stronger network supervision and external supervision. It can effectively restrain the occurrence of opportunistic behavior and help improve the performance of corporate governance.

Based on this, the following hypotheses are proposed in this paper:

**H1:** Digital transformation contributes to improved ESG performance of manufacturing SOEs.

## 2.2 Multiple mechanisms of digital transformation in the perspective of dynamic capabilities

According to the resource-based theory, digital transformation brings enterprises valuable, scarce, and unrepeatable digital resources, including digital technology and data information, which support the practice of corporate social responsibility and become an important source of their competitive advantage. However, in the context of the era of great change, static resources do not guarantee the continuity of the competitive advantage of enterprises, and enterprises need to integrate static resources so that the resources are constantly updated and matched, as a result, dynamic capabilities are formed, which can bring enterprises a constant stream of competitive advantage [34]. Wang and Ahmed divided dynamic capacity into three dimensions: absorptive capacity, adaptive capacity, and innovation capacity, which was recognized by most scholars in China [35,36]. Among them, absorptive capacity refers to the ability of an enterprise to identify, absorb, and transform external information based on its existing knowledge base and apply it to business practices; adaptive capacity refers to the ability of an enterprise to quickly identify and grasp market opportunities and quickly respond to the reallocation of resources; and innovation capacity refers to the ability of an enterprise to develop new products and open new markets. Existing studies have concluded that absorptive, adaptive, and innovative capabilities, as capabilities that can change the conventional capabilities of an enterprise, are crucial in the process of digital transformation [37] and can also play a positive role in promoting specific social responsibility actions. Accordingly, based on the transformation path of "resource-capability-performance", this paper shifts the research perspective to the transformation of multi-dimensional dynamic capabilities triggered by digital transformation and reveals the impact mechanism of digital transformation on corporate ESG performance.

The first is an absorptive feedback mechanism. In the era of the digital economy, the boundaries between enterprises and between enterprises and the outside gradually dissolve, and the application of digital information technology and digital platforms makes the efficiency of enterprises' access to knowledge and information increase and the cost decrease. In the process of digital transformation, enterprises can further improve their ability to identify and acquire internal and external knowledge or information by purchasing new equipment, recruiting highly skilled personnel, and hiring senior consultants [38]. In addition, the contact and communication between enterprises and external stakeholders have become more frequent, and the feedback information from stakeholders helps enterprises deepen their understanding of external needs, purposefully utilize, and improve technological conditions, and further carry out independent research and development to create new knowledge [39]. Based on this, internal knowledge updating and external knowledge sharing in digital enterprises promote resource adjustment and value mining, thus providing opportunities for enterprises to enhance their absorptive capacity. Moreover, this good learning and feedback capability enables digital enterprises to accelerate the absorption and application of advanced manufacturing technologies and management experience at home and abroad, learn lessons in

practice through testing and evaluation, collection of stakeholder feedback, and other channels, iterate their products and governance structures, continuously improve the fulfillment of their environmental, social, and governance responsibilities, and thus improve their ESG performance.

Second, the matching response mechanism. Digital transformation has led to fundamental changes in the internal management mode of enterprises, such as flattening and networking of organizational structure, customization and iterative product design, and modularization and flexibilization of production modes [32], which have enhanced the flexibility and creativity of enterprises so that they can quickly match with the highly dynamic and changing external environment [40], and further enhance the adaptive capacity of enterprises. Enterprises with good adaptive capacity can more keenly capture the changes in the demand for social responsibility and realize the importance and urgency of social responsibility issues. With the help of digital technology, enterprises can understand the changes in the needs of internal and external stakeholders for enterprises by constantly monitoring the social environment and policy trends. Based on the quick response to these changes, enterprises can improve governance efficiency and protect against potential risks, safeguard shareholders' rights and sustainable growth of profits, and at the same time adjust their strategies and goals promptly to better satisfy social needs, seize development opportunities, and assume corresponding social responsibilities, thereby enhancing their ESG performance.

Thirdly, it is an innovative efficiency enhancement mechanism. On the one hand, enterprises need a lot of capital investment to carry out innovative activities, and it is often difficult for endogenous resources to meet the demand, while the application of digital technology can help enterprises improve the quality of information disclosure and attract external investment, thus reducing the cost of enterprise financing and kicking off the financing constraints of this "stumbling block". On the other hand, the use of digital technology can help enterprises identify unutilized valuable resources and find more opportunities for innovation to generate new products or services while reducing the risk of innovation due to information asymmetry [41]. In addition, digital transformation can strengthen the user's demand orientation and timely understanding of changes in consumer demand through online feedback, which in turn improves the enterprise's innovation efficiency and ability [42]. Having good innovation capability enables companies to improve their ESG performance through responsible innovation and green technology innovation based on stakeholders' demands, reducing pollution emissions through innovative production and manufacturing technologies, and improving corporate reputation by providing more environmentally friendly, sustainable, and socially friendly products and services.

Based on this, the following research hypotheses are proposed in this paper:

**H2a:** Digital transformation can enhance firms' ESG performance through the absorptive feedback mechanism, i.e., absorptive capacity mediates the relationship between digital transformation and ESG performance.

**H2b:** Digital transformation can enhance firms' ESG performance by the matching response mechanism, i.e., adaptive capacity mediates the relationship between digital transformation and ESG performance.

**H2c:** Digital transformation can enhance firms' ESG performance through the innovation efficiency enhancement mechanism, i.e., innovation capability mediates the relationship between digital transformation and ESG performance.

## 2.3 Interactive governance mechanisms for digital transformation from an institutional environment perspective

Unlike the stakeholder perspective of CSR, institutional theory emphasizes that the external institutional environment influences firms' performance of social responsibility. Therefore, the difference between the formal and informal institutional environments in which firms are located may affect the relationship between digital transformation and firms' ESG performance.

The formal institutional environment usually includes the market-oriented environment and the legal system, which can reflect the overall status of government intervention, the development of the non-state economy, the product market, the factor market, and the rule of law environment. On the one hand, the better the market-oriented environment, the more transparent enterprises must be in the process of market operation [43], which can reduce the risk of opportunism arising from information asymmetry among enterprises. Moreover, marketization can weaken the adverse effects of government intervention and administrative monopoly, and improve the circulation and allocation efficiency of factors, which significantly reduces the cost of enterprises in acquiring and allocating resources and is conducive to innovation. On the other hand, enterprises carrying out digital transformation often encounter problems such as intellectual property rights in the digital economy and privacy and security of data and information, and a sound legal system can provide strong protection for enterprises to overcome the relevant problems. Therefore, in regions where the formal system is more perfect, the digital transformation of enterprises may realize the synergistic effect of 1+1>2 on the improvement of ESG performance. Based on this, this paper proposes the following research hypotheses:

**H3:** Formal institutions and digital transformation have a synergistic governance effect on firms' ESG performance, i.e., formal institutions produce a positive moderating effect between firms' digital transformation and firms' ESG performance.

In the transitional economic environment, informal institutional arrangements such as social norms, customs, and relationships play the role of modifying, supplementing, or expanding formal rules, and tend to have a stronger binding effect on the responsible behavior choices of enterprises. A large number of practices and researches have shown that informal institutions can influence corporate responsibility fulfillment, such as social trust [43], which can reduce the uncertainty of interactive transactions and enhance the willingness of enterprises to fulfill their social responsibilities; hometown identity [44], which can influence corporate managers' attitudes towards the local environment and make them pay more attention to the needs of the local environmental stakeholders, and enhance the performance of the corporate environmental governance, and so on. Media attention, as the core content of the informal system, also plays an important role in promoting economic and social operations. Especially in the era of the digital economy, the role of online media in social life is growing, and the huge browsing volume and faster dissemination speed make online media reports on social responsibility behaviors greatly stimulate the nerves of enterprises. Due to the lack of owners, government intervention, and less competitive pressure faced, there is a serious principal-agent problem in state-owned enterprises, which cannot implement effective supervision and incentives for operators [45,46]. However, for manufacturing SOEs with a higher degree of digitization, the online media, as the intermediary of online information transmission, will bring stronger public pressure with its positive or negative reports, and this unprecedented huge pressure will motivate the executives to make more correct decisions, which will reduce the agency problem to a certain extent. If corporate decision-making focuses on short-term

interests and refuses to fulfill social responsibility or excessively damages the environment, negative reports from online media can inflict heavy damage on corporate reputation, affecting corporate investment and financing and corporate performance, and state-owned enterprises may also be subject to harsher investigations and penalties by state regulators, which can be considered as a more-than-deserved loss. In addition, media attention can help enterprises disclose more detailed liability information, which can further reduce the information asymmetry between digitally transformed enterprises and stakeholders, alleviate financing constraints, and improve the effectiveness of internal control and corporate governance [47], thus improving the ESG performance of enterprises. Based on this, this paper proposes the following research hypotheses:

**H4:** Informal institutions and digital transformation have a synergistic governance effect on firms' ESG performance, i.e., informal institutions have a positive moderating effect between digital transformation and firms' ESG performance.

The specific theoretical model is shown in Fig 1.

## 3 Research design

### 3.1 Sample selection and data sources

In this paper, the state-owned listed enterprises of the manufacturing industry in Shanghai and Shenzhen A-shares in 2016–2021 are selected as the initial research sample to construct the unbalanced panel model. In this paper, enterprise ESG performance data comes from Huazheng ESG ratings, digital transformation data comes from the annual reports of enterprises on the official websites of the Shanghai Stock Exchange and Shenzhen Stock Exchange, and formal system data comes from China Sub-Provincial Marketization Index 2021. Informal system data comes from the CNRDS database, and all other enterprise-level data comes from the CSMAR database. This paper matches the data according to the security code, year information, etc., and excludes the samples of companies with more missing data values, the

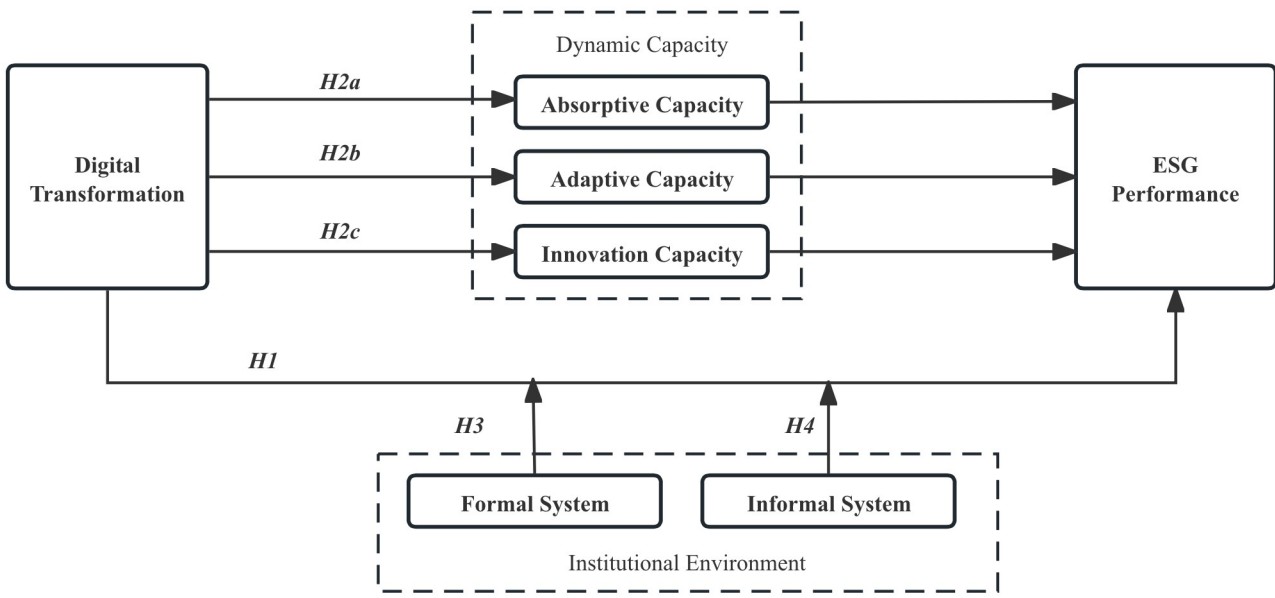

**Fig 1. Theoretical model diagram.**

samples of S.T. companies with abnormal financial conditions, and the samples of companies that were delisted during the period, and finally obtains 3,051 samples. To reduce the interference of outliers, this paper shrinks the observations of continuous variables according to the upper and lower 1% quartiles, and the data processing and model estimation is completed using Python2.7, Excel 2013, and Stata16.0.

## 3.2 Definition of variables

**3.2.1 Explained variable.** Corporate ESG performance (*ESG*): At present, the indicators and caliber of ESG disclosure of Chinese listed companies are different, and the data cannot be compared horizontally, so the feasibility of constructing ESG indicators by oneself is low. The Huazheng ESG ratings incorporate more indicators that fit the current development stage in China and are more suitable for research in the Chinese context, so this paper adopts the results of the Huazheng ESG ratings to measure corporate ESG performance. According to the Huazheng ESG rating, AAA is assigned 9, AA is assigned 8, A is assigned 7, BBB is assigned 6, BB is assigned 5, B is assigned 4, CCC is assigned 3, CC is assigned 2, and C is assigned 1. At the same time, the quarterly ratings are taken as the average value to measure the annual ESG performance, and the higher the ratings are, the better the annual ESG performance of the enterprise.

**3.2.2 Explanatory variable.** Digital Transformation (*DT*): Words about the application of digital technology and digital business scenarios in the annual reports of enterprises can reflect the strategic characteristics and future outlook of the enterprises, and to a large extent reflect the level and development path of the digital transformation of the enterprises, so it can be measured by the word frequency of the corresponding keywords in the annual reports published by the listed enterprises, which can be used as a proxy indicator for the degree of the digital transformation of the enterprises. This paper draws on the research of Wu Fei et al. [48] to construct a digital transformation feature thesaurus, including a total of 76 feature words in terms of artificial intelligence, blockchain, cloud computing, big data, and the application of digital technology. Based on the Shanghai Stock Exchange and Shenzhen Stock Exchange websites to collect and organize the annual reports of sample companies from 2016–2021, use Python 2.7 to extract the text of the annual reports of listed companies to form a data pool, and search, match, and count the word frequency to the feature word thesaurus, and finally add up the total word frequency and then add 1 and do the logarithmic processing, which is used to characterize the digital transformation of enterprises.

**3.2.3 Control variables.** Based on the literature related to digital transformation and ESG, this paper selects corporate age (*Age*), corporate size (*Size*), return on assets (*Roa*), gearing ratio (*Lev*), equity concentration (*Top1*), and two-job integration (*Dual*) indicators as control variables, and classifies the sample companies into subsectors based on the manufacturing industry level 2 of the "Guidelines on Industry Classification of Listed Companies (2012)" of the Securities and Futures Commission. The classification criteria of the sample company's segmented industries are categorized to control for the time effect at the same time. The specific definitions and measures of the main variables are shown in Table 1.

## 3.3 Benchmark modeling

Referring to the practice of related literature [19], this paper adopts OLS (Ordinary Least Squares) to construct the model for multiple regression analysis. To reduce the effect of heteroskedasticity, this paper is analyzed using robust standard errors in the regression.

**Table 1. Definitions of main variables.**

| Variable Type | Variable Name | Variable Symbol | Variable Measurement |
| --- | --- | --- | --- |
| Explanatory Variable | Digital Transformation | *DT* | Natural logarithm of word frequency plus one for words related to digital transformation from text analysis |
| Explained Variable | ESG Performance | *ESG* | Huazheng ESG ratings are averaged for each quarterly score |
| Mechanism Variables | Absorptive Capacity | *DC_Ab* | Company annual R&D expenditure/revenue |
| | Adaptive Capacity | *DC_Ad* | Coefficients of Variation of Annual R&D Expenditures, Capital Expenditures, and Selling Expenses of the Company |
| | Innovation Capacity | *DC_In* | Normalized sum of the company's annual R&D expenditure intensity and technician ratio |
| | Formal System | *Market* | China Sub-Provincial Marketization Index Report 2021 Marketization Index |
| | Informal System | *Media* | The total number of times the company was covered by online media in the year plus 1 taking the natural logarithm |
| Control Variables | Age of Enterprise | *Age* | Current year—year of establishment + 1 in natural logarithms |
| | Enterprise Size | *Size* | Natural logarithm of total assets for the year |
| | Total Return on Assets | *Roa* | Net profit/average balance of total assets |
| | Asset-liability Ratio | *Lev* | Total liabilities at year-end/total assets at year-end |
| | Shareholding Concentration | *Top1* | Shareholding ratio of the largest shareholder |
| | Two Jobs in One | *Dual* | Dummy variable, assigned a value of 1 if the chairman and general manager are the same person, otherwise 0 |
| | Industry Effect | *Industry* | Industry dummy variable, assigned a value of 1 when it belongs to the industry, and 0 otherwise. |
| | Time Effect | *Year* | Year dummy variable, assigned a value of 1 when it belongs to the year and 0 otherwise |

To test the hypothesis H1 proposed in the previous section to construct the benchmark Model (1), the specific regression model is:

$$ESG = \alpha_0 + \alpha_1 DT + \sum \alpha_j Controls + \sum Industry + \sum Year + \varepsilon \qquad (1)$$

where corporate ESG performance (*ESG*) is the explanatory variable, digital transformation (*DT*) is the core explanatory variable and ∑*Controls* denotes control variables, and ∑*Industry* denotes industry fixed effects, and ∑*Year* denotes time fixed effects. Controlling for industry, time, and other control variables, the coefficient $\alpha_1$ characterizes the effect of digital transformation on firms' ESG performance, and if the coefficient $\alpha_1$ is significantly positive, the hypothesis H1 is supported, that is, digital transformation is conducive to improving the ESG performance of manufacturing state-owned enterprises.

## 3.4 Mechanism testing model setting

**3.4.1 Modeling of intermediation mechanisms based on the dynamic capability's perspective.** In order to test whether digital transformation can act on ESG performance through three transmission mechanisms: absorption feedback, matching response, and enhancing innovation efficiency, we used Wen Zhonglin's three-stage mediation effect test method, and constructed models (2) and (3) based on model (1). Model (2) represents the effect of digital transformation on the mediating mechanism variable. Model (3) simultaneously incorporates the core explanatory variables and the mediating mechanism variable into the regression model to examine the direct effect of digital transformation on corporate

ESG performance after the mediating effect is added:

$$M = \alpha_0 + \alpha_1 DT + \sum \alpha_j Controls + \sum Industry + \sum Year + \varepsilon \qquad (2)$$

$$ESG = \alpha_0 + \alpha_1 DT + \alpha_2 M + \sum \alpha_j Controls + \sum Industry + \sum Year + \varepsilon \qquad (3)$$

Among them, $M$ refers to the mechanism variables that digital transformation affects enterprise ESG performance from the perspective of dynamic capability, including absorptive capacity, adaptive capacity, and innovation capacity [35]. The specific measurement methods are as follows:

① Absorptive capacity ($DC\_Ab$): measured using R&D expenditure intensity, i.e., the ratio of annual R&D expenditures to operating revenues of the sample companies.

② Adaptive capacity ($DC\_Ad$): the coefficient of variation of the three major expenditures of R&D expenditures, capital expenditures, and selling expenses of the sample company is used to reflect the flexibility of the enterprise's resource allocation, and then measure the adaptive capacity of the enterprise. To make the value of the coefficient of variation consistent with the adaptive capacity of the direction of the coefficient of variation to take a negative value, the greater the value of the coefficient of variation after adjustment, the stronger the adaptive capacity of the enterprise, the expression formula (where $\sigma$ is the standard deviation of R&D expenditures, capital expenditures, and selling expenses, $mean$ is the average value of the three kinds of expenditures):

$$DC\_Ad = -\frac{\sigma}{mean} \qquad (4)$$

③Innovation capability ($DC\_In$): two indicators of annual R&D expenditure intensity ($RD$) and proportion of technical personnel ($IT$) of the sample companies are used for a comprehensive evaluation, and the data of these two indicators are standardized separately and then summed up to get the comprehensive value of innovation capability:

$$DT\_In = \frac{X_{RD} - min_{RD}}{max_{RD} - min_{RD}} + \frac{X_{IT} - min_{IT}}{max_{IT} - min_{IT}} \qquad (5)$$

Further, absorptive capacity, adaptive capacity, and innovation capacity are standardized and summed to measure the combined utility of dynamic capabilities ($DC$).

**3.4.2 Modeling of the regulatory mechanism based on the institutional environment perspective.** This paper further introduces the moderating terms of formal institutions ($Market$) and informal institutions ($Media$), and constructs models (6) and (7) based on the hierarchical regression method, to clarify the synergistic governance mechanism of the institutional environment and digital transformation on the ESG performance of enterprises.

$$ESG = \beta_0 + \beta_1 DT + \beta_2 Market + \beta_3 DT*Market + \sum \beta_j Controls + \sum Industry + \sum Year + \varepsilon \quad (6)$$

$$ESG = \gamma_0 + \gamma_1 DT + \gamma_2 Media + \gamma_3 DT*Media + \sum \gamma_j Controls + \sum Industry + \sum Year + \varepsilon \quad (7)$$

Among them, the formal system ($Market$) adopts the marketization index in China's Sub-Provincial Marketization Index (2021), which can represent the overall status of the

institutional construction of the product market, the factor market, and the rule of law environment [43]. The higher the score represents the higher the degree of marketization, the better the marketization environment, and the more perfect the corresponding formal system. Secondly, media attention is chosen as an indicator of an informal system (*Media*), which is due to the digital economy, online media, and public opinion reports can often have an unnoticeable impact on the audience and social practice, and the more media reports the higher the degree of influence, the specific measurement method is the total number of times listed companies have been reported by the online media in the year, plus 1 and do logarithmic processing.

## 4 Hypothesis testing and discussion of results

### 4.1 Descriptive statistical analysis

Table 2 presents the descriptive statistics of the main variables of the sample firms. The mean value of *ESG* is 4.1730, the maximum value is 6.2500, the minimum value is 1.2500, and the standard deviation is 1.0440, which indicates that there is a large difference in the ESG performance of the sample enterprises compared with all the Shanghai and Shenzhen A-share-listed enterprises, the state-owned manufacturing industry has a poor ESG performance. The mean value of *DT* is 1.3405, the maximum value is 4.8520, the minimum value is 0.0000, and the standard deviation is 1.2788, indicating that there are large differences in the degree of digital transformation of the sample enterprises, and some enterprises have not yet carried out digital transformation, which may lead to differences in the level of corporate social responsibility, environment and corporate governance, and thus helps to examine the different degrees of digital transformation under the ESG performance of enterprises. The maximum value of *DC* is 15.5231 and the minimum value is -5.8410, with a standard deviation of 2.2726, indicating that there is a large difference in the level of holding dynamic capability among the enterprises in the sample. The minimum value of the *Market* is 4.2100, the maximum value is 11.6900, and the standard deviation is 1.9248, which indicates that there is a large difference in the formal system in the region where the sample firms are located. The minimum value of the *Media* is 2.9444, the maximum value is 8.3561, and the standard deviation is 1.0397, which indicates that there is a large difference in the degree of constraints imposed by the informal system of

**Table 2. Results of descriptive statistics for the main variables.**

| Variables | N | Mean | S.D. | Min | Median | Max |
|---|---|---|---|---|---|---|
| ESG | 3051 | 4.1730 | 1.0440 | 1.2500 | 4.0000 | 6.2500 |
| DT | 3051 | 1.3405 | 1.2439 | 0.0000 | 1.0986 | 4.8520 |
| DC_Ab | 3051 | 0.0407 | 0.0309 | 0.0007 | 0.0357 | 0.1604 |
| DC_Ad | 3051 | -0.7225 | 0.3447 | -1.5539 | -0.6923 | -0.0950 |
| DC_In | 3051 | 0.1498 | 0.1042 | 0.0072 | 0.1306 | 0.5206 |
| DC | 3051 | 0.0000 | 2.2726 | -5.8410 | -0.0797 | 15.5231 |
| Market | 2972 | 8.5666 | 1.9248 | 4.1000 | 8.5300 | 11.6900 |
| Media | 3041 | 5.3536 | 1.0397 | 2.9444 | 5.2311 | 8.3561 |
| Age | 3051 | 3.0574 | 0.2420 | 2.3026 | 3.0910 | 3.5553 |
| Size | 3051 | 22.3802 | 1.2881 | 19.6770 | 22.3110 | 25.9293 |
| Roa | 3051 | 0.0361 | 0.0566 | -0.1580 | 0.0309 | 0.2068 |
| Lev | 3051 | 0.4506 | 0.1910 | 0.0752 | 0.4552 | 0.8965 |
| Top1 | 3051 | 0.3538 | 0.1389 | 0.0974 | 0.3308 | 0.7280 |
| Dual | 3051 | 0.1308 | 0.3372 | 0.0000 | 0.0000 | 1.0000 |

the sample firms. The standard deviations of all other control variables are large, indicating that their observed values vary significantly across firms, which in turn may affect firms' ESG performance. In addition, after the covariance diagnosis, the values of VIF for the above variables are less than 10, thus there is no more serious covariance relationship.

## 4.2 Benchmark model testing

To test the effect of digital transformation on the ESG performance of manufacturing state-owned enterprises, this paper conducts regression analysis based on the benchmark model (1), and the test results are detailed in Table 3. Column 1 in Table 3 shows the regression results after adding the relevant control variables and the industry effect and time effect. The results show that the regression coefficient of digital transformation ($DT$) is 0.0809 and is significantly positive at a 1% confidence level. It indicates that digital transformation is conducive to improving the ESG performance of manufacturing state-owned enterprises, and every 1% increase in the degree of digital transformation will promote the ESG performance of enterprises by 0.0809%, and hypothesis H1 is verified.

There may be endogeneity issues between digital transformation and ESG performance, so to ensure the robustness of the findings this paper further conducts the following tests. First, to control the endogeneity problem caused by bidirectional causality, this paper re-tests the explanatory variables with a 1-period lag, and the regression results are shown in Table 3, Column 2. Second, to control the endogeneity problem caused by omitted variables, this paper

**Table 3. Benchmark regression analysis and endogeneity test.**

| Variables | 1 | 2 | 3 | 4 | 5 | 6 |
|---|---|---|---|---|---|---|
| | *ESG* | *ESG$_{t+1}$* | *ESG* | *PSM_nem* | *PSM_cm* | *PSM_km* |
| DT | 0.0809*** | 0.0700*** | 0.0779*** | 0.2204*** | 0.1504*** | 0.1504*** |
| | (4.67) | (3.52) | (4.53) | (4.16) | (3.81) | (3.81) |
| Age | -0.1645** | -0.1666* | -0.1666** | -0.1940* | -0.1499* | -0.1499* |
| | (-2.00) | (-1.69) | (-2.02) | (-1.67) | (-1.82) | (-1.82) |
| Size | 0.0552*** | 0.0625*** | 0.0552*** | 0.0612*** | 0.0533*** | 0.0533*** |
| | (3.83) | (3.79) | (3.79) | (3.28) | (3.70) | (3.70) |
| Roa | 4.2000*** | 4.6129*** | 4.3562*** | 4.8493*** | 4.2844*** | 4.2844*** |
| | (10.80) | (10.50) | (11.68) | (9.50) | (10.90) | (10.90) |
| Lev | -0.0682 | -0.0249 | -0.0320 | -0.0532 | -0.0440 | -0.0440 |
| | (-0.60) | (-0.19) | (-0.29) | (-0.34) | (-0.39) | (-0.39) |
| Top1 | 0.5789*** | 0.6001*** | 0.5643*** | 0.5452*** | 0.5672*** | 0.5672*** |
| | (4.21) | (3.85) | (4.03) | (2.81) | (4.11) | (4.11) |
| Dual | -0.2604*** | -0.2614*** | -0.2717*** | -0.1740** | -0.2556*** | -0.2556*** |
| | (-5.02) | (-4.32) | (-4.91) | (-2.37) | (-4.94) | (-4.94) |
| Industry Effect | Yes | Yes | Yes | Yes | Yes | Yes |
| Time Effect | Yes | Yes | Yes | Yes | Yes | Yes |
| Industry x Time Joint Effect | No | No | Yes | No | No | No |
| _cons | 2.2813*** | 2.6310*** | 3.0347*** | 2.1180*** | 2.2612*** | 2.2612*** |
| | (5.68) | (5.37) | (7.54) | (3.77) | (5.62) | (5.62) |
| N | 3051 | 2423 | 3020 | 1680 | 3046 | 3046 |
| Adj.$^2$ | 0.1252 | 0.1286 | 0.1046 | 0.1299 | 0.1232 | 0.1232 |

Note1: ***, **, * indicate statistical tests at the 1%, 5%, and 10% significance levels respectively.

Note2: Yes/No represents controlled/uncontrolled, the same below.

further re-tests the model by adding the industry x time joint fixed effects, and the regression results are shown in Table 3, Column 3. Third, to control the endogeneity problem caused by sample self-selection, this paper further uses propensity score matching (PSM) to estimate the net effect of digital transformation on firms' ESG performance. In this paper, samples smaller than or equal to the median of digital transformation are used as the control group, samples higher than the median of digital transformation are used as the treatment group, and the control variables above are selected as the covariates. The methods of one-to-one nearest neighbor matching, caliper matching (caliper = 0.05), and kernel matching were used respectively to match the treatment group and the control group. The T-values of the average treatment effect (ATT) were 4.35, 3.99, and 3.95, respectively, and the regression was conducted using the matched samples after passing the balance test, the regression results are detailed in Columns 4 to 6 of Table 3. In the above test results, the regression coefficients of the digital transformation (*DT*) are all significantly positive at the 1% confidence level, and the core conclusions of the paper are robust.

## 4.3 Analysis of the marginal effect of digital transformation at different ESG performance levels

The above results indicate that digital transformation is an important factor in promoting the improvement of ESG performance of state-owned manufacturing enterprises, but the impact on the average level is discussed, which fails to answer such questions. That is, when the ESG performance of enterprises is at different levels, what is the difference in the marginal impact of digital transformation? To answer this question, the paper further employs panel quantile regression. This paper selects five quartiles of 10%, 25%, 50%, 75% & 90% for regression. The test results are shown in Table 4, the digital transformation (*DT*) regression coefficients increase with the increase of ESG quartiles, and there are obvious structural differences in the impact of corporate digital transformation on ESG performance. The regression coefficients of digital transformation (*DT*) for the 10% and 25% quartiles are insignificant, which suggests that the digital transformation will mainly drive up the ESG performance of the middle and high levels. This implies that firms with high ESG performance are more positively affected in the process of digital transformation than firms with poor ESG performance and that digital transformation may cause a widening gap in ESG performance among firms.

This may be because firms with better ESG performance, especially manufacturing firms, are often perceived to manage environmental issues better than their peers, and analysts and investors direct and accelerate external capital flows to such firms [49], making them more resilient in the face of long term risks, and thus the goals of digital transformation for such firms are more inclined to be on how to utilize digital technologies to improve their social responsibility performance, corporate reputation, and sustainability. Companies with lower levels of ESG performance have lower positioning of their products or services and lower awareness of responsibility fulfillment. The goal of digital transformation for such enterprises is to reduce costs and improve operational efficiency and performance in the short term, resulting in an insignificant impact on ESG performance.

## 4.4 Mechanism studies

**4.4.1 Examination of multiple mechanisms for digital transformation in a dynamic capability's perspective.** Based on the conduction path of "resource-capability-performance", this paper adopts the stepwise regression method according to model (2) and model (3) to test the absorptive feedback, matching response and innovation efficiency enhancement mechanism of digital transformation, and the test results are shown in Table 5a to 5d.

**Table 4. Analysis of marginal effects.**

| | 1 | 2 | 3 | 4 | 5 |
|---|---|---|---|---|---|
| | *P10* | *P25* | *P50* | *P75* | *P90* |
| DT | -0.0016 | 0.0290 | 0.0949** | 0.1194*** | 0.1701*** |
| | (-0.04) | (0.86) | (2.35) | (3.47) | (3.13) |
| Age | -0.1668 | -0.0030 | -0.0310 | -0.0239 | 0.2423* |
| | (-1.15) | (-0.02) | (-0.15) | (-0.14) | (1.95) |
| Size | 0.0520 | 0.0621 | 0.0783*** | 0.0467** | 0.0253 |
| | (1.02) | (1.62) | (3.27) | (2.11) | (1.09) |
| Roa | 4.7438*** | 4.2970*** | 4.8334*** | 4.1507*** | 3.8799*** |
| | (9.59) | (8.47) | (4.92) | (6.91) | (6.09) |
| Lev | -0.3135 | -0.3327 | 0.0285 | 0.1743 | 0.3814 |
| | (-1.20) | (-1.42) | (0.13) | (0.56) | (1.14) |
| Top1 | 1.4352*** | 0.8307*** | 0.4933 | 0.2207 | 0.3640 |
| | (3.88) | (2.63) | (1.12) | (0.81) | (1.29) |
| Dual | -0.4757*** | -0.2879* | -0.1079 | -0.2371*** | -0.2661*** |
| | (-4.12) | (-1.91) | (-1.34) | (-4.51) | (-3.91) |
| Industry Effect | Yes | Yes | Yes | Yes | Yes |
| Time Effect | Yes | Yes | Yes | Yes | Yes |
| N | 3051 | 3051 | 3051 | 3051 | 3051 |

Table 5a reports the results of testing the absorption feedback mechanism of digital transformation. Column 1 presents the regression results of digital transformation (*DT*) on absorptive capacity (*DC_Ab*), where the regression coefficient of digital transformation (*DT*) is significantly positive at the 1% level, suggesting that digital transformation contributes to the firm's absorptive capacity. Column 2 shows the results of the full-variable regression, in which the regression coefficients of digital transformation (*DT*) and absorptive capacity (*DC_Ab*) are significantly positive at the 1% and 5% levels, respectively, and the coefficient of digital transformation (*DT*) is reduced from 0.0809 to 0.0765, which suggests that absorptive capacity has a partially mediating role in the relationship between digital transformation and firms' ESG performances, i.e., digital transformation can improve firms' ESG performances through the absorptive feedback mechanism to improve firms' ESG performance, hypothesizing that H2a holds.

Table 5b reports the results of testing the matching response mechanism of digital transformation. Column 1 presents the regression results of digital transformation (*DT*) on adaptive capacity (*DC_Ad*), where the regression coefficient of digital transformation (*DT*) is significantly positive at the 1% level, suggesting that digital transformation contributes to the adaptive capacity of firms. Column 2 shows the results of all-variable regression, in which the regression coefficients of digital transformation (*DT*) and adaptive capacity (*DC_Ad*) are both significantly positive at the 1% level, and the coefficient of digital transformation (*DT*) is reduced from 0.0809 to 0.0775, indicating that adaptive capacity has a partially mediating role in the relationship between digital transformation and firms' ESG performance, i.e., digital transformation can improve firms' ESG performance through the matching response mechanism by improve firms' ESG performance, assuming that H2b holds.

Table 5c reports the results of testing the innovation efficiency enhancement mechanism of digital transformation. Column 1 presents the regression results of digital transformation (*DT*) on innovation capability (*DC_In*), where the regression coefficient of innovation capability (*DC_In*) is significantly positive at the 1% level, indicating that digital transformation

**Table 5.** a. Absorption feedback mechanism test. b. Matching response mechanism test. c. Innovation efficiency enhancement mechanism test. d. Dynamic capacity synthesis mechanism test.

| | 1 | 2 |
|---|---|---|
| | DC_Ab | ESG |
| DT | 0.0025*** | 0.0765*** |
| | (5.07) | (4.39) |
| DC_Ab | | 1.7375** |
| | | (2.43) |
| Controls | Yes | Yes |
| Industry Effect | Yes | Yes |
| Time Effect | Yes | Yes |
| _cons | 0.0930*** | 2.1197*** |
| | (8.92) | (5.16) |
| N | 3051 | 3051 |
| Adj.² | 0.3812 | 0.1266 |
| | 1 | 2 |
| | DC_Ad | ESG |
| DT | 0.0218*** | 0.0775*** |
| | (4.05) | (4.48) |
| DC_Ad | | 0.1571*** |
| | | (2.66) |
| Controls | Yes | Yes |
| Industry Effect | Yes | Yes |
| Time Effect | Yes | Yes |
| _cons | -0.5722*** | 2.3712*** |
| | (-4.37) | (5.90) |
| N | 3051 | 3051 |
| Adj.² | 0.1882 | 0.1271 |
| | 1 | 2 |
| | DC_In | ESG |
| DT | 0.0177*** | 0.0717*** |
| | (9.63) | (4.05) |
| DC_In | | 0.5233** |
| | | (2.53) |
| Controls | Yes | Yes |
| Industry Effect | Yes | Yes |
| Time Effect | Yes | Yes |
| _cons | 0.2404*** | 2.1555*** |
| | (6.94) | (5.34) |
| N | 3051 | 3051 |
| Adj.² | 0.3223 | 0.1268 |
| | 2 | 3 |
| | DC | ESG |
| DT | 0.3087*** | 0.0697*** |
| | (8.60) | (3.98) |
| DC | | 0.0364*** |
| | | (3.69) |
| Controls | Yes | Yes |
| Industry Effect | Yes | Yes |

(*Continued*)

**Table 5.** (Continued)

| | | |
|---|---|---|
| Time Effect | Yes | Yes |
| _cons | 2.8667*** | 2.1771*** |
| | (3.71) | (5.41) |
| N | 3051 | 3051 |
| Adj.$^2$ | 0.3756 | 0.1288 |

contributes to the adaptive capacity of firms. Column 2 shows the full-variable regression results, in which the regression coefficients of digital transformation (*DT*) and innovation capability (*DC_In*) are significantly positive at the 1% and 5% levels, respectively, and the coefficient of digital transformation (*DT*) is reduced from 0.0809 to 0.0717, which suggests that there is a partially mediating role of innovation capability in the relationship between digital transformation and firms' ESG performances, i.e., digital transformation can improve firms' ESG performance by innovation efficiency enhancement mechanism to improve firms' ESG performance, and hypothesis H2c hold.

Further, the combined utility of dynamic capabilities (*DC*) is tested, and the regression results are shown in Table 5d. The regression coefficient of digital transformation (*DT*) in Column 1 is significantly positive at the 1% level, indicating that digital transformation contributes to the improvement of the firm's combined dynamic capabilities. Column 2 shows the full-variable regression results, in which the regression coefficients of digital transformation (*DT*) and dynamic capability (*DC*) are both significantly positive at the 1% level, and the coefficient of digital transformation (*DT*) is reduced from 0.0809 to 0.0697, which further verifies the mediating mechanism role of comprehensive dynamic capability. In summary, digital transformation under the perspective of dynamic capabilities can improve the ESG performance of enterprises through the triple mechanism of absorption feedback, matching response, and innovation efficiency enhancement.

**4.4.2 Examination of interactive governance mechanisms for digital transformation from the perspective of the institutional environment.** To clarify the relationship between institutional environment and digital transformation on the ESG performance of enterprises, this paper examines the synergistic governance mechanism of formal and informal institutions according to the models (6) to (7) respectively, and the test results are detailed in Table 6.

Column 1 of Table 6 examines the effect of formal institutions on firms' ESG performance, and it can be seen that the regression coefficient of formal institutions (*Market*) is 0.0560, which is significantly positive at the 1% level, which indicates that the more complete the formal institutions are, the better the firms' ESG performance is; after both digital transformation (*DT*) and the interaction term of digital transformation and formal institutions (*DT*Market*) are added in Column 3 the regression coefficient of digital transformation (*DT*) and formal system (*Market*) is still significantly positive, but the regression coefficient of the interaction term of digital transformation and formal system (*DT*Market*) is negative (-0.0055) and insignificant, and the hypothesis H3 is not valid. This may be because the regional factor market with a more complete formal system is relatively developed, and the motivation of state-owned enterprises to use digital transformation to obtain external resources will be slightly insufficient. In addition, as the degree of market transparency continues to improve due to institutional development, state-owned enterprises will also be strongly impacted by other enterprises in the industry. In the context of relatively limited total resources, state-owned enterprises tend to maintain established economic benefits and business objectives, so digital transformation may reduce their investment in social responsibility [50]. Moreover, when

**Table 6. Institutional environmental governance mechanisms test.**

|  | 1 | 2 | 3 | 4 | 5 |
|---|---|---|---|---|---|
|  | *ESG* | *ESG* | *ESG* | *ESG* | *ESG* |
| *DT* |  |  | 0.0698*** | 0.0517*** | 0.0411** |
|  |  |  | (3.92) | (2.93) | (2.26) |
| *Market* | 0.0560*** |  | 0.0525*** |  | 0.0495*** |
|  | (5.38) |  | (5.03) |  | (4.79) |
| *Media* |  | 0.1503*** |  | 0.1224*** | 0.1223*** |
|  |  | (7.92) |  | (6.37) | (6.34) |
| *DT\* Market* |  |  | -0.0055 |  | -0.0054 |
|  |  |  | (-0.62) |  | (-0.63) |
| *DT\* Media* |  |  |  | 0.0684*** | 0.0694*** |
|  |  |  |  | (5.09) | (5.07) |
| Controls | Yes | Yes | Yes | Yes | Yes |
| Industry Effect | Yes | Yes | Yes | Yes | Yes |
| Time Effect | Yes | Yes | Yes | Yes | Yes |
| _cons | 1.9203*** | 1.5101*** | 2.5112*** | 2.3916*** | 2.4959*** |
|  | (4.71) | (3.67) | (6.10) | (5.96) | (6.09) |
| N | 2972 | 3041 | 2972 | 3041 | 2962 |
| Adj.$^2$ | 0.1276 | 0.1347 | 0.1318 | 0.1439 | 0.1504 |

enterprises are in a good legal environment, the marginal role of digital transformation may be exceeded by the supervision role of the legal system on corporate responsibility behavior, that is, the fulfillment of individual responsibilities will be more dependent on the formal system, thus dilution of the positive effect of digital transformation on the ESG performance of enterprises.

Column 2 of Table 6 examines the effect of informal system on firms' ESG performance, which shows that the regression coefficient of informal system (*Media*) is 0.1503, which is significantly positive at 1% level, which indicates that the stronger the constraints of the informal system, the better the firm's ESG performance is; Column 4 incorporates both the digital transformation (*DT*) as well as the interaction term between the digital transformation and the informal system (*DT\*Media*), the interaction term of digital transformation and informal system (*DT\*Media*) is significantly positive at the 1% level (0.0684), which indicates that the effect of firms' digital transformation based on advancing corporate ESG responsibility practices is stronger when the pressure of informal systems on firms is stronger, and that digitalization improves the transparency of firms' information, expands the effect of media governance, and helps to multiply the role of informal systems in promoting the corporate ESG performance, i.e., there is a synergistic governance effect between digital transformation and informal systems, and hypothesis H4 is tested.

Further, the above variables were put into the model at the same time, and the test results are shown in Column 5 of Table 6, where the significance and direction of the coefficients of the interaction terms did not change, which once again verified that the informal institution in the institutional environment has a synergistic governance effect with the digital transformation, whereas the formal institution does not have a significant synergistic governance effect with the digital transformation.

**Table 7. Robustness tests.**

| | 1 | 2 | 3 | 4 | 5 | 6 | 7 |
|---|---|---|---|---|---|---|---|
| | *E* | *S* | *G* | *ESG* | *DC* | *ESG* | *ESG* |
| *DT* | 1.5034*** | 0.9352*** | 0.6605*** | | | | |
| | (4.65) | (5.68) | (4.00) | | | | |
| *DTmd&a* | | | | 0.1335*** | 0.7847*** | 0.0987*** | 0.0683* |
| | | | | (3.73) | (10.50) | (2.72) | (1.79) |
| *DC* | | | | | | 0.0444*** | |
| | | | | | | (3.86) | |
| *DTmd&a\*Market* | | | | | | | -0.0063 |
| | | | | | | | (-0.35) |
| *DTmd&a\*Media* | | | | | | | 0.0915*** |
| | | | | | | | (3.81) |
| Controls | Yes | Yes | Yes | Yes | Yes | Yes | Yes |
| Industry Effect | Yes | Yes | Yes | Yes | Yes | Yes | Yes |
| Time Effect | Yes | Yes | Yes | Yes | Yes | Yes | Yes |
| _cons | -54.7730*** | -10.5706** | 68.5090*** | 2.6467*** | 2.1347** | 2.5520*** | 2.8225*** |
| | (-6.30) | (-2.24) | (15.55) | (5.62) | (2.44) | (5.40) | (5.86) |
| *N* | 1455 | 1455 | 1455 | 2357 | 2357 | 2357 | 2281 |
| Adj.$^2$ | 0.2780 | 0.1828 | 0.2306 | 0.1104 | 0.3956 | 0.1156 | 0.1371 |

## 4.5 Robustness tests

To further validate the robustness of the findings, the following robustness tests were conducted. First, splitting the ESG dimensions: re-examining the impact of digital transformation on the three-segmented dimensions E, S, and G. The results of the regression are shown in Columns 1 to 7 of Table 7. The test results are shown in columns 1 to 3 of Table 7, and the significance and direction of the regression results are not fundamentally changed. Digitalization has the largest positive impact on the E performance dimension, followed by the S performance dimension, and the G performance dimension. Second, replace the explanatory variables: Referring to the study of Yuan Chun et al. (2021), the degree of digitization of micro-enterprises is measured by dividing the sum of the frequency of digitization-related words by the length of the "Management's Discussion and Analysis" (MD&A) section of the annual report [51]. All the models above are re-tested and the results are shown in Columns 4 to 7 of Table 7. The significance and direction of the regression results have not been fundamentally altered, and the above conclusions are still robust. Third, for the mediation effect test, some scholars believe that the direct test of the coefficient product method is better than the stepwise regression method, to guarantee the reliability of the results, this paper adopts the Bootstrap method to replace the stepwise regression method to re-test the test results are shown in Table 8. The bias-corrected confidence intervals for the indirect effects of the mechanism variables do not contain 0, i.e., there are partially mediated effects of the mechanism variables of the dynamic capabilities, and the conclusions are still robust.

## 5. Discussion

The empirical results are as follows. First, digital transformation is conducive to improving the ESG performance of manufacturing SOEs. This conclusion still holds after a series of endogeneity tests. Further quantile tests find that there are significant structural differences in the impact of digital transformation on ESG performance, with the marginal impact effect of

**Table 8. Robustness test of mediating effect.**

| Variable | Direct effect | Indirect effect | The proportion of indirect effects | Confidence interval | Conclusion |
|---|---|---|---|---|---|
| DC_Ab | 0.0765 | 0.0044 | 5.47% | [0.0013, 0.0111] | partial mediation |
| DC_Ad | 0.0775 | 0.0034 | 4.23% | [0.0016, 0.0075] | partial mediation |
| DC_In | 0.0717 | 0.0112 | 11.44% | [0.0016, 0.0075] | partial mediation |
| DC | 0.0697 | 0.0112 | 13.87% | [0.0016, 0.0075] | partial mediation |

digital transformation increasing as firms' ESG performance improves. The reason may be that analysts and investors guide and accelerate the flow of external capital to firms with better ESG performance [49], widening the gap in ESG performance between firms. Second, under the dynamic capability's perspective, absorptive capacity, adaptive capacity, and innovation capacity are mediating variables that facilitate the transformation of digital resources into ESG performance. Absorptive capacity enables firms to better absorb external knowledge to improve products, governance structure, etc., and to form feedback to stakeholders' needs, so the application of digital resources can improve the absorptive capacity of firms and be transformed into the ESG performance of firms through the absorptive feedback mechanism. Adaptive capacity enables firms to quickly match the highly dynamic and changing external environment and respond to changes in stakeholder needs, so the application of digital resources improves the adaptive capacity of firms and further translates into the ESG performance of firms through the matching response mechanism. The improvement of innovation capability unexpectedly leads to the improvement of innovation resource allocation efficiency, and enterprises can innovate according to stakeholders' needs at lower cost and higher quality, so the application of digital resources can improve the innovation capability of enterprises and further be transformed into the ESG performance of enterprises through the mechanism of innovation efficiency enhancement. Third, from the institutional environment perspective, informal institutions are moderating variables that affect digital transformation and ESG performance. Digitalization improves the transparency of corporate information and expands the effect of media governance, so digital transformation helps to multiply the role of informal institutions in promoting corporate ESG performance, and there is a synergistic governance effect between the two. The moderating effect of formal institutions on digital transformation and ESG performance is not significant. The reason may be that the more developed the formal system is, the more transparent the market is, and the SOEs are subject to strong impacts from other firms in the industry. In the context of relatively limited total resources, SOEs tend to maintain the established economic efficiency and business objectives, and thus undergoing digital transformation may scale down their investment in social responsibility. Moreover, when firms are in a better rule-of-law environment, the marginal effect of digital transformation may be outweighed by the monitoring effect of the legal system on responsible corporate behavior, thus diluting the positive effect of digital transformation on firms' ESG performance.

The contributions of this study compared to previous studies are as follows. Compared with the literature [21,52], the impact of digital transformation on ESG performance is empirically examined from the perspective of manufacturing SOEs. Compared with the literature [2,19,20], different mechanisms of digital transformation affecting ESG performance are discussed based on the dynamic capability theory and institutional environment theory. Therefore, this study has important theoretical and practical implications, as follows. Firstly, previous literature has few studies on state-owned enterprises (SOEs) and the conclusions are inconsistent, this study clarifies the controversy over the relationship between digital transformation Also, this study enriches the research on the influencing factors of ESG performance of

manufacturing SOEs. Secondly, the study complements the gap in the research on the mechanism of digital transformation affecting ESG performance. It explores the effect of digital transformation from the perspectives of institutional environment and dynamic capability. It opens the "black box" of the impact of digital transformation on the ESG performance of manufacturing SOEs. Thirdly, it provides a theoretical basis for the implementation of digital transformation to improve the ESG performance of manufacturing SOEs. The findings of the study can help manufacturing SOEs to grasp the new opportunities of digital development and become the "front-runner" in ESG construction.

## 6. Conclusions and enlightenments

### 6.1 Conclusions

As a pillar industry of the national economy and an important force in the process of economic and social development, the social responsibility behavior of manufacturing enterprises is of key significance to building a harmonious society and realizing the goal of sustainable development. However, in China, the manufacturing industry is an industry with high ESG risk. State-owned enterprises (SOEs), as the leading enterprises in key manufacturing industries, should be the "front-runners" in the construction of China's ESG system and play a leading role in the fulfillment of other enterprises' social responsibility. Based on this, in order to clarify the influencing factors of ESG performance of manufacturing state-owned enterprises, and to help manufacturing enterprises seize the new opportunities of digital development and improve their ESG performance, this paper takes Shanghai and Shenzhen A-share manufacturing state-owned listed enterprises as the research samples, empirically examines the impact and marginal effect of digital transformation on the ESG performance of the enterprises, and discusses the influencing mechanism of digital transformation from the perspectives of dynamic capability and institutional environment. Discussions are carried out. The results of the study are as follows.

First, digital transformation helps improve the ESG performance of manufacturing SOEs. Second, there are significant structural differences in this effect, with the marginal impact effect of digital transformation increasing as firms' ESG performance improves. Third, there are partial mediating effects of absorptive capacity, adaptive capacity, and innovation capacity in dynamic capabilities. That is, digital transformation can improve corporate ESG performance through the absorptive feedback mechanism, matching response mechanism, and innovation efficiency enhancement mechanism. Fourth, in the institutional environment, the moderating effect of the formal institutional environment on digital transformation and ESG performance is not significant. There is a significant positive moderating effect of informal institutions on the relationship between digital transformation and ESG performance, i.e., informal institutions and digital transformation have a synergistic governance effect on corporate ESG performance.

### 6.2 Management enlightenments

This study also brings some management insights and policy recommendations. First, state-owned enterprises (SOEs) should grasp the new opportunities of the digital economy era, emphasize the incentive role of digital transformation on corporate ESG practices, and do a good job in building China's ESG system while considering their development. In recent years, the ESG performance of enterprises has gradually become an important standard for regulators and investment institutions to measure the potential for sustainable development of enterprises, and the further promotion of digital transformation can have a positive impact on the environment, society, and corporate governance, which is conducive to attracting external investment and realizing the goal of sustainable development. Second, SOEs should emphasize

the cultivation of their dynamic capabilities. In the process of digital transformation, enterprises should not only focus on the digital technology itself, but also focus on cultivating the absorptive, adaptive, and innovative capabilities of the enterprise, actively perceiving changes in the needs of stakeholders, proactively searching for and absorbing new knowledge and innovating the original products and services and accelerating the transformation of digital resources into non-economic performance. Third, practicing ESG is a "must-answer" question for realizing China's sustainable development goals. The government should establish and improve corporate ESG evaluation standards and information disclosure systems as soon as possible, actively publicize ESG concepts, strengthen the awareness of corporate responsibility, and guide corporations to strike a balance between economic benefits and social responsibility in their operations. At the same time, for the digital transformation of corporate responsibility to fulfill the role of incentives, the government should further improve the digital infrastructure and supporting services, strengthen the digital transformation guide, strengthen the government's financial support for the development of enterprise digitalization, to stimulate the enthusiasm of the digital transformation of the traditional manufacturing enterprises, to improve the part of the enterprise "do not want to turn", "dare to turn", "difficult to succeed" situation. Fourthly, the media should play the role of informal institutional regulators in the capital market to monitor the ESG disclosure of enterprises and their social responsibility performance, promote ESG practices, and improve the current status quo of poor ESG performance of manufacturing enterprises.

## 6.3 Limitations

This study has some limitations, which need to be improved in future research. First, although this paper provides a theoretical basis for manufacturing SOEs to implement digital transformation to improve ESG performance, it lacks more detailed guidance and needs to be further supplemented with relevant case studies. Second, there is still room for further deepening the research on the mechanism of digital transformation affecting ESG performance. For example, digital technology plays an important role in supply chain management and industrial structure upgrading [53], which may become an important mechanism for digital transformation to influence the sustainable development of enterprises. Third, the explanatory variables and paths selected in this paper that may have an impact on firms' ESG performance are relatively single, and in the future, multiple possible explanatory variables can be selected from multiple perspectives such as capabilities, resources, and strategies for group analysis to further explore the multiple causal mechanisms that promote the improvement of firms' ESG performance.

## Supporting information

**S1 Data.**
(ZIP)

## Author Contributions

**Conceptualization:** Xin Jin, Yizhu Wu.

**Data curation:** Yizhu Wu.

**Formal analysis:** Xin Jin.

**Methodology:** Yizhu Wu.

**Software:** Yizhu Wu.

**Supervision:** Xin Jin.

**Validation:** Yizhu Wu.

**Writing – original draft:** Yizhu Wu.

**Writing – review & editing:** Xin Jin, Yizhu Wu.

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
