## [Decision Letter · Decision Letter 0]

24 Jan 2024

PONE-D-23-43453How does digital transformation impact corporate ESG performance？ Evidence from China's state-owned manufacturing enterprisesPLOS ONE

Dear Dr. Wu,

Thank you for submitting your manuscript to PLOS ONE. After careful consideration, we feel that it has merit but does not fully meet PLOS ONE’s publication criteria as it currently stands. Therefore, we invite you to submit a revised version of the manuscript that addresses the points raised during the review process.

We look forward to receiving your revised manuscript.

Kind regards,

Pengyu Chen

Academic Editor

PLOS ONE

Journal Requirements:

Reviewers' comments:

Reviewer's Responses to Questions

**Comments to the Author**

1. Is the manuscript technically sound, and do the data support the conclusions?

Reviewer #1: Partly

Reviewer #2: Yes

2. Has the statistical analysis been performed appropriately and rigorously? 

Reviewer #1: Yes

Reviewer #2: Yes

3. Have the authors made all data underlying the findings in their manuscript fully available?

Reviewer #1: No

Reviewer #2: Yes

4. Is the manuscript presented in an intelligible fashion and written in standard English?

Reviewer #1: Yes

Reviewer #2: Yes

5. Review Comments to the Author

Reviewer #1: The reviewer believes that the topic “How does digital transformation impact corporate ESG performance？ Evidence from China's state-owned manufacturing enterprises” is worthy of investigation. However, the following needs to be addressed. There are minor and major issues that should be corrected. I believe the paper could be further strengthened by added information about.

Please reorganize the manuscript at the journal request. Please change the reference format.

The language of this manuscript is very bad and needs help from native speakers.

The title of the manuscript should fully demonstrate the content of this study and the relevant subjects.

Abstracts should include the purpose and findings of the study.

Introduction . This a very vague statement. These sentences do not provide any information on how the concept could be conceptualized?

This section should explain the study's context and research objective. Furthermore, the research gap needs to be narrowed after analyzing the previous studies. The research method is not adequately explained in the first section.

-Introduction, what authors wanted to convey. Here author must build research gap following the previous studies.-The manuscript does not answer the following concerns: Why is it timeliness to explore such a study? What makes this study different from the previously published studies? Are there any similarly findings in line with the previously published studies? Are the findings different from prior academic studies that were conducted elsewhere, if any? For example, information innovation and innovation network, what it requires, what are the new technologies, some recent issue highlights the importance. See the following: Enhancing digital innovation for the sustainable transformation of manufacturing industry: a pressure-state-response system framework to perceptions of digital green innovation and its performance for green and intelligent manufacturing.

Developing a Conceptual Partner Selection Framework: Digital Green Innovation Management of Prefabricated Construction Enterprises for Sustainable Urban Development.

An adoption-implementation framework of digital green knowledge to improve the performance of digital green innovation practices for industry 5.0;

-Methodology: Model.. I suggest authors here build your main heading on Research and data methodology. Clearly explain the model building process, and what previous studies have used similar models (model testing approach).

There is no flow in the text. It partly depends on the lack of proofreading but also on the fact that many statements and claims are made without being followed up by a clear and logical discussion. It is especially problematic in the Introduction that brings up a number of findings from different areas without linking them together.

Please make sure your conclusions' section underscores the scientific value-added of your paper, and/or the applicability of your findings/results. Highlight the novelty of your study.

In addition to summarizing the actions taken and results, please strengthen the explanation of their significance. It is recommended to use quantitative reasoning comparing with appropriate benchmarks, especially those stemming from previous work. See the following:Developing a Conceptual Partner Matching Framework for Digital Green Innovation of Agricultural High-End Equipment Manufacturing System Toward Agriculture 5.0: A Novel Niche Field Model Combined With Fuzzy VIKOR

More importantly, the choice of the variables should be explained in light of the theory and the prior literature on the topic. The arguments are simply relationships and causes very close to the replication of many studies dealing with the same thing.

The authors should emphasize the important role of digital technology in industrial structure upgrading in future research. Some recent issue highlights the importance: The Interaction Mechanism and Dynamic Evolution of Digital Green Innovation in the Integrated Green Building Supply Chain.

Please consider this structure for manuscript final part.

-Discussion

-Conclusion

-Managerial Implication

-Practical/Social Implications

-Discussion needs to be a coherent and cohesive set of arguments that take us beyond this study in particular, and help us see the relevance of what authors have proposed. Authors should create an independent “Discussion” section. Author need to contextualize the findings in the literature, and need to be explicit about the added value of your study towards that literature. Also other studies should be cited to increase the theoretical background of each of the method used. Findings should be contextualized in the literature and should be explicit about the added value of the study towards the literature (New Energy-Driven Construction Industry: Digital Green Innovation Investment Project Selection of Photovoltaic Building Materials Enterprises Using an Integrated Fuzzy Decision Approach). Limitations and future research.

As any emprical study that use different approaches I would like to ask to introduce in the Conclusion section at least a paragraph containing the study limitations. I noticed some things in the paper but a synthesis of statements related to how the study is useful (or partially useful, since are required certain further analysis) and helps potential interested readers does not really exist. Maybe in addition to the last section of Conclusion it is beneficial to introduce a section called: Discussion.

Reviewer #2: Manuscript title: How does digital transformation impact corporate ESG performance？ Evidence from China's state-owned manufacturing enterprises

Manuscript ID:PONE-D-23-43453

Summary of the paper: This paper studies the relationship between digital transformation and ESG performance. Using a sample of Chinese listed firms, the authors document that digital transformation is an important factor in promoting the improvement of ESG performance of state-owned enterprises. The authors further investigate the mechanism of digital transformation from the perspectives of dynamic capabilities and institutional environment. These findings can provide empirical evidence for improving ESG performance and promoting corporate sustainability. The paper concentrates on an important issue about ESG performance in the context of emerging economics or developing countries. In addition, the moderating effects of institutional environment influence ESG practices of enterprises in the long-run. The authors highlight different interactive governance mechanisms for digital transformation and how they influence the ESG practices of Chinese listed firms. This paper examines an interesting question and obtains supporting evidence. However, there are a few comments and concerns on the current draft. I suggest to the authors only minor revisions:

1. Relationship to the literature and contribution:The discussion of contributions lacks reference support. A more nuanced discussion of extant research and their outcomes would be necessary to contextualize the contribution of the paper. For example, in line 94-106: The authors state that ”explores the mechanism of digital transformation from the perspectives of institutional environment and dynamic capability....."Are these mechanism analysis concepts widely used by related literature? The description lacks literature support,which might confuse the first contribution with the second.The paper would benefit greatly if the authors could point out more clearly in the introduction section that how exactly this paper provides novel insights into the literature. 

2. Research hypothesis :Line 145-148:The authors state that “The performance of corporate social responsibility mainly emphasizes that enterprises should strengthen the governance of the relationship between multiple stakeholders....."This may seem applicable to all enterprises(including SOE and non-SOE). The authors should emphasize that this situation is unique to state-owned enterprises and update the whole paragraph.

3.Line 165-186:The authors hypothesize that: “The corporate governance performance of an enterprise includes two dimensions: the internal governance and the external governance performance....”in Section 1.1 . However, moderating role of Formal institutions and Informal institutions are highlighted again in Section 1.3. It will mislead readers to confuse on the real mechanism of digital transformation. In addition, H2 and H3 are proven by moderating effects,which is different from dynamic capabilities mechanism.

4.Theoretical framework:It is recommended to reshape the theoretical framework and add a diagram or figure to help readers quickly understand the whole story and core research variables.

5.Robustness Tests:The robustness test should display the related table on page21.It is very clear that the authors emphasize Digital Transformation have positive effect on Corporate E,S and G in section2. The paper would benefit greatly if the authors could show the different impacts of digital transformation on three aspects of ESG.

6.Other comments: 

The formula in line 428 and 433 is not labeled.

The hypothesis order is incorrect.

There are some grammar mistakes. The author should conduct grammar checks and proofreading

There is a wealth of literature on digital transformation and ESG. The authors should cite English literature rather than Chinese.

6. PLOS authors have the option to publish the peer review history of their article (what does this mean?). If published, this will include your full peer review and any attached files.

Reviewer #1: No

Reviewer #2: No

---

## [Author Response · Author response to Decision Letter 0]

7 Mar 2024

Response to Reviewer #1：

Thank you very much for allowing us to rewrite the paper. We have carefully read and collated your comments and tried our best to rewrite the paper. We appreciate your serious evaluation of our work and thank you for giving us a chance to improve our work.

Comment 1: Please reorganize the manuscript at the journal’s request. Please change the reference format. 

Response 1: We apologize for the formatting error. We downloaded the format of the Endnote references provided by PLoS and updated the citations.

Comment 2: The language of this manuscript is very bad and needs help from native speakers.

Response 2: Regarding grammar, we have re-proofread the lines and used touch-up software to touch up the language of the article in the hope of compensating for the language deficiencies of our non-native speakers.

Comment 3: The title of the manuscript should fully demonstrate the content of this study and the relevant subjects. 

Response 3: Thanks to your suggestion, we have revised the title of the article. This study empirically examines the impact of digital transformation on ESG performance using a sample of Chinese manufacturing state-owned enterprises. It explores the internal transmission mechanism of dynamic capabilities in the relationship between digital transformation and ESG performance. As well as the external regulation mechanism of the institutional environment in the relationship between digital transformation and ESG performance. Therefore, we have incorporated the above research focus into the title of the paper in the hope of fully demonstrating the content and the related subjects of this study. The revised title of the paper is: How does digital transformation affect the ESG performance of Chinese manufacturing state-owned enterprises? --Based on the mediating mechanism of dynamic capabilities and the moderating mechanism of the institutional environment.

Comment 4: Abstracts should include the purpose and findings of the study.

Response 4: Thank you for your suggestions. We have rewritten the abstract section. In the abstract section, we briefly summarized the background of the study, the purpose of the study, the sample and methodology, the test results and findings, and the significance of the study.

Comment 5: The introduction section should explain the study's context and research objective. Furthermore, the research gap needs to be narrowed after analyzing the previous studies. The research method is not adequately explained in the first section. （Here author must build a research gap following the previous studies. The manuscript does not answer the following concerns: Why is it timeliness to explore such a study? What makes this study different from the previously published studies? Are there any similar findings in line with the previously published studies? Are the findings different from prior academic studies that were conducted elsewhere, if any? For example, information innovation and innovation network, what it requires, what are the new technologies, some recent issue highlights the importance.） 

Response 5: Thank you for your suggestions. In response to your questions, we have made several changes to the introduction.

1) We emphasize the importance of improving ESG performance in the context of China's "dual-carbon" policy and the concept of sustainable development, as well as the reality of the poor ESG performance of manufacturing state-owned enterprises (SOEs) (Line 40-63).

2) We add a review of studies related to ESG and digitalization. We provide a review of the factors influencing ESG, the consequences of digital transformation, and the existing mechanisms of how digital transformation affects ESG (Line 89-98).

3) Based on previous studies, we find that although many scholars have demonstrated the existence of a positive impact of digital transformation on ESG performance, there are still shortcomings. On the one hand, few existing studies cut from the perspective of state-owned enterprises (SOEs), and the relationship between digital transformation and ESG performance of manufacturing SOEs may be controversial (Line 116-136). On the other hand, research on the mechanism of digital transformation affecting ESG mostly focuses on internal control, green innovation, financing constraints, etc., and there are many gaps (Line 95-115).

Based on the above realistic background and research gaps, this paper takes Shanghai and Shenzhen A-share state-owned listed enterprises in the manufacturing industry as the research sample and empirically examines the impact of digital transformation on the ESG performance of the sample enterprises using regression analysis. Further attempts to discuss the influence mechanism of digital transformation from the perspectives of dynamic capabilities and the institutional environment through stepwise regression and hierarchical regression methods respectively (Line 137-149).

Comment 6: Methodology: Model. I suggest authors here build your main heading on Research and data methodology. Clearly explain the model building process, and what previous studies have used similar models (model testing approach). 

Response 6: Thank you for your suggestions. We have revised the main title of the article and the subheadings in the model construction chapters to hopefully show more clearly the significance of each model construction. In addition, we have added references where similar models have been used and a description of the model construction methodology (Line 431-432, 449-451, 482-484).

Comment 7: There is no flow in the text. It partly depends on the lack of proofreading but also on the fact that many statements and claims are made without being followed up by a clear and logical discussion. It is especially problematic in the Introduction that brings up several findings from different areas without linking them together. 

Response 7: Thank you very much for your suggestion. We have updated the introduction section (See Response 5 for details). The article is based on the research based on the real-world problem highlighted in the updated introduction, i.e., what can be done to improve the poor ESG performance of state-owned manufacturing industries in the context of sustainable development policies? The selection of variables is also based on the research gap emphasized in the updated introduction. In addition, we have proofread the discussion before and after the paper in the hope of providing readers with a clearer understanding of the points we want to make.

Comment 8: Please make sure your conclusions section underscores the scientific value-added of your paper, and/or the applicability of your findings/results. Highlight the novelty of your study. In addition to summarizing the actions taken and results, please strengthen the explanation of their significance. It is recommended to use quantitative reasoning compared with appropriate benchmarks, especially those stemming from previous work.

Response 8: Thank you very much for your advice. We drew on the paper you recommended and enhanced the comparison between the contents of this study and previous studies in the discussion section (Line 751-755) to highlight the possible contributions of this study.

Comment 9: As with any empirical study that uses different approaches, I would like to ask you to introduce in the Conclusion section at least a paragraph containing the study’s limitations. Moreover, the authors should emphasize the important role of digital technology in industrial structure upgrading in future research. Some recent issue highlights the importance of the Interaction Mechanism and Dynamic Evolution of Digital Green Innovation in the Integrated Green Building Supply Chain. 

Response 9: Thank you for your advice. In the last part of the conclusion (6.3), we introduce the limitations of this study and the prospect of future research. With reference to the literature recommended by you, we point out that the mechanism of digital transformation affecting the sustainable development of enterprises still needs to be deepened in future studies, and further emphasize the important role of digital technology in supply chain management and industrial structure upgrading (Line 833-837).

Comment 10:

Please consider this structure for manuscript’s final part.

-Discussion

-Conclusion

-Managerial Implication

-Practical/Social Implications

The discussion needs to be a coherent and cohesive set of arguments that take us beyond this study and help us see the relevance of what the authors have proposed. Authors should create an independent “Discussion” section. Findings should be contextualized in the literature and should be explicit about the added value of the study towards the literature. 

Response 10: Thank you very much for your advice. We combined your suggestions with some of the articles you recommended (The Interaction Mechanism and Dynamic Evolution of Digital Green Innovation in the Integrated Green Building Supply Chain) and changed the structure of the last part of the article. We created a separate "discussion" section. In this part, we summarize the research arguments and test results. In addition, based on the comparison with the existing literature, we summarized the possible theoretical and practical significance of this study. In the conclusion part, we summarize the research background and conclusion. Based on the research conclusions, the paper gives some relevant management implications and policy recommendations. Finally, the limitations of the study and the prospect of future research are introduced.

Response to Reviewer #2：

Thank you very much for giving us a chance to rewrite our paper. We have read the comments to the referees carefully and tried our best to rewrite the paper. We appreciate your positive comments on our work and it’s kind of you to give us a chance to perfect our work. 

Comment 1: Relationship to the literature and contribution: The discussion of contributions lacks reference support. A more nuanced discussion of extant research and their outcomes would be necessary to contextualize the contribution of the paper. For example, in line 94-106: The authors state that” explores the mechanism of digital transformation from the perspectives of institutional environment and dynamic capability....."Are these mechanism analysis concepts widely used by related literature? The description lacks literature support, which might confuse the first contribution with the second. The paper would benefit greatly if the authors could point out more clearly in the introduction section that how exactly this paper provides novel insights into the literature. 

Response 1: Thank you very much for your advice. We have supplemented the introduction with an extensive review of research on digital transformation, ESG, and its impact mechanisms. Based on the review of existing studies, we point out that: 

1) the internal management decisions and external regulatory policies of enterprises are the focus of early ESG influencing factors research (Line 64-78). In recent years, many studies have highlighted the potential of digital technologies to improve sustainability performance, as well as the positive impact of digitization on ESG performance (Line 79-95). 

2) Although studies have demonstrated the positive impact of digital transformation on ESG performance, there are still shortcomings. On the one hand, few existing studies start from the perspective of state-owned enterprises, and the relationship between digital transformation and the ESG performance of state-owned enterprises in manufacturing may be controversial (Line 116-136). On the other hand, research on the mechanism of digital transformation affecting ESG mostly focuses on internal control, green innovation, financing constraints, etc., and there are many gaps (Line 95-115).

Comment 2: Research hypothesis: Line 145-148: The authors state that “The performance of corporate social responsibility mainly emphasizes that enterprises should strengthen the governance of the relationship between multiple stakeholders....."This may seem applicable to all enterprises(including SOE and non-SOE). The authors should emphasize that this situation is unique to state-owned enterprises and update the whole paragraph.

Response 2: Thanks for your suggestion, this statement is intended to explain to the reader the specific definition of S in ESG. However, after your reminder, we found that such a statement may cause misunderstanding to readers. Moreover, the whole paragraph does lack a discussion of the specific situation of state-owned enterprises. Therefore, we have updated the whole paragraph to point out the practical problem that SOEs pay less attention to external social responsibilities related to external stakeholders (Line 196-201). Digitization can enable traditional manufacturing state-owned enterprises to identify and respond more quickly to the demands of external stakeholders and improve the above problems. In addition, it can further improve the performance of responsibility of state-owned enterprises to internal stakeholders (Line 199-212).

In addition, we also checked other paragraphs for similar problems and supplemented the relevant discussion (eg. Line 659-666).

Comment 3: Line 165-186: The authors hypothesize that: “The corporate governance performance of an enterprise includes two dimensions: the internal governance and the external governance performance....”in Section 1.1. However, moderating role of Formal institutions and Informal institutions are highlighted again in Section 1.3. It will mislead readers to confuse on the real mechanism of digital transformation. In addition, H2 and H3 are proven by moderating effects, which is different from dynamic capabilities mechanism. 

Response 3: Thank you very much for your suggestion. The intent of the paragraph in section 1.1 was to discuss the impact of digital transformation on the corporate governance performance (G) of manufacturing SOEs, however, it may have created a misunderstanding for the reader. We have updated the discussion in that paragraph in the hope that it can be distinguished from mechanism studies.

In addition, we apologize for the mislabeling of assumptions in the text that led to misunderstanding by readers. We have corrected the assumption labeling throughout the text. Among them, H1 is the hypothesis of the direct effect of digital transformation, H2a~H2c are the hypotheses of the dynamic capability mediation mechanism, and H3 and H4 are the hypotheses of the institutional environment regulation mechanism.

Comment 4: Theoretical framework: It is recommended to reshape the theoretical framework and add a diagram or figure to help readers quickly understand the whole story and core research variables.

Response 4: We think this is a good suggestion. We have added a model diagram containing the research hypotheses at the end of the Theoretical analysis and research hypothesis chapter (Fig. 1) in the hope that it will give the reader a clearer understanding of what we have done in our research.

Comment 5: Robustness Tests: The robustness test should display the related table on page 21. It is very clear that the authors emphasize Digital Transformation has a positive effect on Corporate E, S, and G in section 2. The paper would benefit greatly if the authors could show the different impacts of digital transformation on three aspects of ESG. 

Response 5: Thanks to your suggestion, we have supplemented the table with the results of the robustness tests. Moreover, we have added the test of the impact of digital transformation on the E, S, and G subdimensions in the robustness test section. The test results show that Digitalization has the largest positive impact on the E performance dimension, followed by the S performance dimension, and the G performance dimension. 

Comment 6: Other comments: 

The formula in line 428 and 433 is not labeled.

The hypothesis order is incorrect.

There are some grammar mistakes. The author should conduct grammar checks and proofreading

There is a wealth of literature on digital transformation and ESG. The authors should cite English literature rather 

---

## [Decision Letter · Decision Letter 1]

25 Mar 2024

How does digital transformation affect the ESG performance of Chinese manufacturing state-owned enterprises? --Based on the mediating mechanism of dynamic capabilities and the moderating mechanism of the institutional environment

PONE-D-23-43453R1

Dear Dr. Wu,

We’re pleased to inform you that your manuscript has been judged scientifically suitable for publication and will be formally accepted for publication once it meets all outstanding technical requirements.

Kind regards,

Pengyu Chen

Academic Editor

PLOS ONE

Additional Editor Comments (optional):

Reviewers' comments:

Reviewer's Responses to Questions

**Comments to the Author**

1. If the authors have adequately addressed your comments raised in a previous round of review and you feel that this manuscript is now acceptable for publication, you may indicate that here to bypass the “Comments to the Author” section, enter your conflict of interest statement in the “Confidential to Editor” section, and submit your "Accept" recommendation.

Reviewer #1: (No Response)

Reviewer #2: All comments have been addressed

2. Is the manuscript technically sound, and do the data support the conclusions?

Reviewer #1: (No Response)

Reviewer #2: Yes

3. Has the statistical analysis been performed appropriately and rigorously? 

Reviewer #1: (No Response)

Reviewer #2: Yes

4. Have the authors made all data underlying the findings in their manuscript fully available?

Reviewer #1: (No Response)

Reviewer #2: Yes

5. Is the manuscript presented in an intelligible fashion and written in standard English?

Reviewer #1: (No Response)

Reviewer #2: Yes

6. Review Comments to the Author

Reviewer #1: I am satisfied with the revisions carried out based on earlier feedback. The paper is in need of a final language check, preferably by an experienced or professional proofreader, to improve the clarity of expression and impact of your ideas. Once this is resolved, your paper will be ready for acceptance.

Reviewer #2: The authors have adequately addressed the comments raised in a previous round of review and I feel that this manuscript is now acceptable for publication. If time permits，please confirm whether the formula in the full text is displayed correctly.

7. PLOS authors have the option to publish the peer review history of their article (what does this mean?). If published, this will include your full peer review and any attached files.

Reviewer #1: No

Reviewer #2: No

---

## [Editor Report · Acceptance letter]

2 May 2024

PONE-D-23-43453R1 

PLOS ONE

Dear Dr. Wu, 

I'm pleased to inform you that your manuscript has been deemed suitable for publication in PLOS ONE. Congratulations! Your manuscript is now being handed over to our production team.

Kind regards, 

on behalf of

Dr. Pengyu Chen 

Academic Editor

PLOS ONE